# The Effects of Pregnancy: A Systematic Review of Adolescent Pregnancy in Ghana, Liberia, and Nigeria

**DOI:** 10.3390/ijerph20010605

**Published:** 2022-12-29

**Authors:** Augustine Lambonmung, Charity Asantewaa Acheampong, Uma Langkulsen

**Affiliations:** 1Faculty of Public Health, Thammasat University, Pathum Thani 12120, Thailand; 2Tamale Teaching Hospital, Ministry of Health, Tamale P.O. Box TL 16, Ghana; 3Princess Marie Louise (PML) Children Hospital, Ghana Health Service, Accra P.O. Box GP 122, Ghana

**Keywords:** adolescent pregnancy, health and well-being, SDG 3, West Africa

## Abstract

There is a high incidence of adolescent pregnancy in West Africa. The objective of this study is to highlight the health impacts of adolescent pregnancy through a systematic review. A search was conducted in the electronic databases of Google, Google Scholar, SCOPUS, EBSCO, CINAHL, Web of Science, African Journals Online (AJOL), and the Demographic Health Surveys (DHS) Program. The study found anemia, complications of pregnancy, obstetric and gynecological risks, unsafe abortions, and psychological effects to adversely impact the health of adolescent girls in Ghana, Liberia, and Nigeria. Pregnancy could be deleterious to the health and well-being of adolescent girls in various forms. In addition, adolescent pregnancy could expose adolescent girls to gender-based violence, exclusions, and inequities, be detrimental to upholding women’s sexual and reproductive health rights, and could also have implications for Sustainable Development Goal 3. Targeted interventions to prevent pregnancy in young women and mitigate these effects by stakeholders are encouraged.

## 1. Introduction

The adolescence (10–19) years are a transitional period between childhood and adulthood [1]. There are about 1.2 billion adolescents globally with over 50% of them from Asia. Sub-Saharan Africa has the largest proportion of their general population [2]. For this group of young people to grow and make meaningful contributions to society, their fundamental human rights must be ensured and they must be well-educated. The provision of jobs, inclusiveness, and important issues relating to their health should effectively be dealt with [3]. Adolescents are faced with myriad health challenges with an increased risk depending on which part of the world they find themselves in [4]. Sexual and reproductive health and rights among adolescents have been an ongoing global health concern given the fact that they are regarded as critical for the progress of society.

The issue of adolescent health as a global health challenge is due to the complexity of meeting sexual and reproductive health needs and the provision of antenatal care (ANC) services to adolescents. It requires a multi-sectoral and collaborative approach to prevent of unwanted pregnancies and alleviate the negative consequences of adolescent pregnancy [5]. The World Health Organization (WHO) issued 18 guidelines in 2011 seeking to prevent adolescent pregnancies and improve reproductive health in Low- and Middle-Income Countries (LMICs) [6]. In spite of this and other local interventions, pregnancies among young girls in LMICs are unacceptably high. According to the World Health Organization (WHO), about 21 million girls (15–19) years become mothers and a further 2 million younger than 15 years old give birth annually across the world, with most of them occurring in LMICs, specifically countries in South Asia and Sub-Saharan Africa. Further, three million pregnancies are unsafely terminated which can negatively impact the health of these girls [7]. Adolescent pregnancy in West Africa, as in other regions of the developing world, is among the highest in the world [8]. The rate of adolescent pregnancy in the sub-region has been persistently high, with a prevalence of about 25% spanning a period of two decades (1992–2011) [9]. An adolescent birth rate of 115 per 1000 births in West Africa in recent times is considered the highest of any sub-region globally [2]. Moreover, pregnancy at a tender age could adversely affect young girls diversely. Pregnant adolescent girls in LMICs are impacted by pregnancy in many facets of their lives, particularly in health, employment, education, and to some extent, family members later in life [10]. Though there is a decline in mortality and morbidity of children and adolescents globally (from 1990–2015), it is sadly observed that maternal and reproductive health-related issues are significantly contributing to the disease burden of adolescent girls in LMICs [4]. Three broad determining factors for adolescent pregnancy contributing significantly to this chronic public health concern were identified as socio-cultural, environmental and economic, and individual and health services [11]. This global health challenge, which has adverse consequences for birth outcomes and poverty eradication and can transverse generations [12], remains high in West Africa. Issues such as poverty, weak health systems, gender inequality, and some cultural practices, such as early marriages, are implicated in adolescent pregnancy [8]. The highly restrictive abortion regimes in most countries in the region with their clandestine abortion practices further injure the health of adolescent girls [13]. The health impacts of pregnancy on adolescent girls in Ghana, Liberia, and Nigeria in West Africa have been examined by a few studies. The three countries purposively selected are English-speaking countries since publications in the English language only were considered for the study. They are representative of the similarities in the region with regard to politics, low socioeconomic development, and geography, as well as demographic backgrounds. Liberia is a country emerging from a devastating civil war and has a very high rate of child marriage, a low level of educated girls, and a high incidence of adolescent pregnancy in Sub-Saharan Africa. Nigeria, the most populated nation in West Africa, is considered one of the countries with the highest rate of adolescent pregnancy and a low level of educated girls globally [14]. Ghana has been stable politically and economically among its peers in the sub-region. With about 11% of their respective populations being adolescent girls (10–19) years, they have high adolescent birth rates. Ghana, with 78 per 1000 girls (15–19) years, is among the lowest in the West Africa region, followed by Nigeria with 106 and Liberia with 128, which is considered among the highest worldwide [2].

Highlighting how pregnancy is defiling the health of adolescent girls in these countries in West Africa through a systematic review could help in adopting more pragmatic approaches to check unplanned pregnancies, especially in younger adolescents, and institute measures to improve adolescent maternal health outcomes to further reduce maternal mortality in line with the Millennium Development Goals (MDGs) to help achieve the Sustainable Development Goals (SDG 3). SDG 3 and targets 3.1 and 3.7, which are closely linked to the objectives of this review, seek to reduce the maternal mortality ratio and ensure wider coverage of sexual and reproductive health care services, respectively [15]. The study would also be beneficial in recognizing the need for the fulfillment of sexual and reproductive health, and the rights of women, especially safe abortion care services without undue legal restrictions. This study intends to bring together the maternal health impacts of adolescent pregnancy in the context of West Africa with a focus on Ghana, Liberia, and Nigeria.

## 2. Materials and Methods

### 2.1. Study Design and Search Strategy

The design for the study is a systematic review. A review protocol was developed and registered in PROSPERO (CRD42021289636). Preferred Reporting Items for Systematic Reviews and Meta-Analysis (PRISMA) 2020 guidelines [16] were adhered to. Google, Google Scholar, SCOPUS, EBSCO, CINAHL, Web of Science, African Journals Online (AJOL), and the Demographic and Health Surveys (DHS) Program were searched in November 2021 using the following terms: adolescent pregnancy, adolescent birth, and teenage pregnancy. Search terms such as outcomes, risks, impacts, effects, consequences, and associated factors were also used. 

### 2.2. Selection Criteria

A broad search framework was constructed and implemented for a thorough online search of applicable studies. The inclusion criteria applied for the purposes of gathering literature for the review were publications and articles published from the year 2016, post-Millennium Development Goals (MDGs), to 2021 were considered. Abstracts were screened, and a full text was retrieved and assessed for eligibility. Publications that discussed adolescent pregnancies in the selected countries were reviewed. Reports and publications by organizations (governmental and non-governmental bodies) on the impacts of adolescent pregnancy in the selected West African nations were evaluated for inclusion. Publications in the English language relating to the search terms relevant for the review and analyzed the effects of pregnancy in adolescent women between the ages of 10–19 years, determined to be useful for the review, were used. Publications were excluded from the review if they lacked relevant data on pregnant adolescents aged 10–19 years. Studies on adolescent pregnancy conducted outside of the region were not considered for the review. 

### 2.3. Data Extraction and Quality Assessment

The Joanna Briggs appraisal tool was independently used by two reviewers to ascertain their quality for inclusion or exclusion. The tool is a checklist made up of a set of eleven questions [17]. Where there was a disagreement or due to inadequate information, the full text of the disputed abstract was obtained, and a third reviewer mediated following a discussion on the issue for a consensus to be reached. Studies that reported different outcomes from the same study sample were excluded for the likelihood of publication bias. The latest and final reports of the relevant Demographic and Health Surveys were included. For a possible publication bias, some articles were excluded from the final synthesis of the review [18,19,20]. 

### 2.4. Data Synthesis

A narrative synthesis was chosen to carefully select the clearest reported outcomes that were directly relevant to the review question and were grouped under common themes suitable to the objective of the study. The narrative synthesis complied with the recommended criteria in the performance of a rigorous synthesis without meta-analysis (SWiM) [21]. This provides a clear guide for reporting on the evaluation of studies using alternative synthesis approaches to a meta-analysis of effect estimates. It allowed for the extracted data to be summarized, structured, and described independently of the reviewers, and a consensus formed over disputed results. The effects of pregnancy on girls were combined and homogenously grouped as anemia, complications of pregnancy and obstetric and gynecological effects, unsafe abortions, psychological effects based on conceptual appropriateness, methods employed, and how unbiased they were in terms of designing, conducting, and analyzing the studies. 

## 3. Results

The 28 studies included in the review were conducted in Ghana, Liberia, and Nigeria in the West African sub-region, and were used after they were blindly reviewed by two independent reviewers and a third reviewer whenever there was no consensus between independent reviewers. Figure 1 illustrates the screening steps for inclusion and exclusion. Three studies that reported different outcomes from the same study were excluded from the final synthesis of results for possible publication bias.

### 3.1. Characteristics of Included Studies 

As shown in Table 1, most of the studies used were qualitative, cross-sectional studies, and surveys (five each). Three of the surveys were demographic and health surveys (final reports) and two were national surveys. There were three retrospective studies. Two of the studies used descriptive, prospective, and longitudinal designs. One of the two longitudinal studies used Adolescent Birth Outcomes, Ghana (ANBOG). There was one randomized control trial, exploratory, observational, and mixed study design (qualitative and quantitative) studies each for the reviews.

Participants in the studies used had at least a sample size of 17 [22] and the highest sample size was 26,055 [23]. These studies were carried out in both rural and urban settings, and were institution-based and community-based. A total of 47,479 adolescent girls eligible for the studies were pregnant either during or before the studies were undertaken. Their ages ranged from 11 years (youngest) to 24 years (oldest) and included educated and non-educated, and married and unmarried. Some key informants comprising parents, teachers, adolescent mothers and partners, and health care providers participated in some of the qualitative studies used, providing useful and relevant information on how pregnancy impacts the health of adolescent girls.

It is worthy of note that none of the studies included in this review included studies on vulnerable adolescent groups, such as the disabled or those living with non-communicable diseases (NCDs) or victims of pandemics, such as HIV/AIDS or Ebola, which have devastated the sub-region. There was no study found to have investigated COVID-19-positive pregnant adolescent girls in the selected countries in West Africa. Detailed characteristics of all included studies are summarized in Table 1.

**Table 1 ijerph-20-00605-t001:** Characteristics of included studies.

Author, Year	Setting and Country	Participants	Sample Size	Age Range of Participants (Year)	Education	Employment Status	Marital Status	Gestational Age (Week)	Residence	Antenatal Care Attendance
Adeniyi et al., 2021 [24]	Federal teaching hospital, Ido, Ekiti, Nigeria	Pregnant teenagers	116 (58 cases and 58 controls)	14–19	Secondary and tertiary = 48.2% No education = 1.7%	Earn income = 8.6%	Unmarried = 82.8% Married = 8.6%	N/A	N/A	Yes = 32.8% No = 67.2%
Ampiah et al., 2019 [25]	7 district health centers in the Ashanti region, Ghana	Pregnant teenagers	998 (119 teenagers)	13–19	N/A	N/A	N/A	Up to 36 weeks	N/A	Yes = All
Annan et al., 2021 [26]	29 communities in Kumasi metropolis, Ghana	Pregnant teenagers	416	13–19	N/A	N/A	N/A	Up to 32 weeks	urban	Yes
Annan et al., 2021 [20]	Health center based in Ashanti, Ghana	Pregnant adolescents	416	13–19	None = 4.6%	Unemployed = 71.6% Employed = 28.4%	Married = 24% Single = 76%	Up to 32 weeks	N/A	N/A
Appiah et al., 2021 [27]	Ledzorkuku-Krowor in Greater Accra, Ghana	Pregnant adolescents	423	12–19 (16–19 = 71.4%)	No education = 16.5%	Petty trading = 45.2%	Married = 9%	Up to 32 weeks	N/A	Yes
Ayamolowo et al., 2019 [28]	Osun State, South-West, Nigeria	Pregnant and child-rearing teenagers	120	13–20	Non = 11.7% Primary and above = 88.3%	N/A	Married = 46.7% Single = 50%	N/A	N/A	N/A
Engelbert et al., 2019 [29]	Jamestown Accra, Ghana	Adolescents (30)	53	14–19	Students = 3	Unemployed = 17	Single = 27	N/A	N/A	N/A
Bain et al., 2020 [30]	Jamestown, Accra, Ghana	Adolescents, parents, and teachers	54	N/A	N/A	N/A	N/A	N/A	N/A	N/A
Dare et al., 2016 [31]	Angwan Rukuba, Jos, Plateau State, Nigeria	33 teenagers (pregnant or mothers) and 67 were never pregnant	100	13–18 (13–14 = 38, 15–16 = 42, 17–18 = 20)	Primary education = 60%, No education =2%	Unemployed = 7% Street hawkers = 30%	N/A	N/A	N/A	N/A
Envuladu et al., 2017 [32]	2 local government areas of Plateau State, Nigeria	Adolescents (males and females), teachers, and healthcare providers	24	18–19 (adolescents)	N/A	N/A	N/A	N/A	N/A	N/A
Gbogbo, 2020 [33]	Hohoe municipality, Ghana	Adolescents (mothers, pregnant)	92	15–19	Basic school dropout = 77	Employed = 2 Unemployed = 90	Married = 4 Unmarried = 88	N/A	Rural = 9	N/A
Ghana 2017 DHS, 2018 [34]	National, Ghana	Pregnant adolescents	365	<20	N/A	N/A	N/A	N/A	N/A	N/A
Gyimah et al., 2021 [18]	Ashanti region, Ghana	Pregnant adolescents	416	13–19	Only basic education 61.3%	Employed = 28.4% Unemployed = 71.6%	Married = 24% Single = 76%	16 weeks average	Urban = 58.4% Rural = 41.6%	N/A
Gyimah et al., 2021 [19]	29 districts of Kumasi Metropolis, Ghana	Pregnant adolescents	416	13–19	Junior High School = 61.3%	Unemployed = 71.6%	Unmarried = 76.0%	Up to 32 weeks	N/A	Yes
Keogh et al., 2021 [35]	National, Ghana	Adolescents	4139 (1039 adolescents)	15–24	N/A	N/A	N/A	N/A	N/A	N/A
Konneh et al., 2020 [36]	Jackson Doe Referral Hospital, Liberia	Adolescents and adult mothers	1265 (540, 43% adolescents)	11–19	Primary = 65.9%	N/A	N/A	N/A	N/A	N/A
Krugu et al., 2017 [37]	Bolgatanga municipality, Ghana	Young women with pregnancy experience	20	14–19	School dropouts = 11	Socioeconomic status not indicated	Married = 7 Single = 13	N/A	N/A	N/A
Kuyinu et al., 2020 [38]	Lagos Island, Nigeria	Pregnant adolescents	246	16–19 (adolescent girls)	N/A	N/A	N/A	1st–3rd trimester	N/A	1st–3rd trimester
Liberia 2019–2020 DHS, 2021 [39]	National, Liberia	Pregnant adolescents	213	<20	N/A	N/A	N/A	N/A	N/A	N/A
Nigeria 2018 DHS, 2019 [14]	National, Nigeria	Pregnant adolescents	47	<20	N/A	N/A	N/A	N/A	N/A	N/A
Nonterah et al., 2019 [40]	Navarongo War Memorial Hospital, Ghana	Pregnant women	506 (33 adolescents)	< 20	N/A	N/A	N/A	5–36 weeks	N/A	Yes
Oladeji et al., 2019 [41]	Ibadan, Southwest, Nigeria	Pregnant adolescents and adults	9352 (772 adolescents)	<19	N/A	N/A	Married or cohabiting = 53.4%	21.3 weeks	N/A	N/A
Olajubu et al., 2021 [42]	Ile-Ife, Osun State, Nigeria	Pregnant teenagers	241	14–19	Educated = 90%	No source of income = 64.3%	Married = 32.8% Single= 67.2%	N/A	N/A	N/A
Olorunsaiye et al., 2021 [22]	Jos, Plateau state, Nigeria	Adolescents and young women with experience of adolescent pregnancy	17	16–24	All completed at least Junior High School	Employed = 41.2% Unemployed = 17.6%	Never married = 82.3% Separated or divorced = 17.7%	N/A	N/A	N/A
Oyeyemi et al., 2019 [43]	Maiduguri, capital of Borno state, Nigeria	Young mothers	220 (110 adolescents)	14–17	No education = 75.8%	Farmers = 54.5% Traders = 50.7%	Married = 51.1% Unmarried = 40%	N/A	N/A	Yes = 47.4% No = 59.2%
Siakwa et al., 2020 [44]	3 hospitals in Cape Coast Metropolis, Ghana	Pregnant teenagers	1006 (503 adolescents)	13–19	No education = 16.0% Basic education = 65.9%	Employed = 33.7% Unemployed = 66.3%	Married = 16.8% Single = 78.2%	18 weeks average	Urban	Yes = 88%
Tetteh et al., 2020 [23]	Nigeria	Teenagers with pregnancy experience	26,055 (Nigeria = 8423)	15–19	Non = 25.8% Primary and above = 74.2%	N/A	Married = 22.8% Not married = 77.2%	N/A	Urban = 45.1% Rural = 54.9%	N/A
Yussif et al., 2017 [45]	A community in Northern Ghana	Women	143 (46 adolescents)	<19	No education = 40	Traders = 90%	Married = 93%	N/A	N/A	N/A

As summarized in Table 2, this review found four main themes on how pregnancy affects the health and well-being of adolescent girls in West Africa. They are anemia, complications of pregnancy, obstetric and gynecological effects, unsafe abortions, and psychological effects.

### 3.2. Anaemia 

Anemia as determined by the six studies is a common complication that occurs among pregnant adolescents in these countries. Higher anemia among adolescents [24], iron deficiency anemia common among pregnant teenagers [25], the presence of morbidity in the form of anemia [26], and adolescent expectant mothers more likely to be anemic [27] were the findings under this theme.

### 3.3. Pregnancy-Related Complications and Obstetric and Gynecological Effects

The various unpleasant health effects during pregnancy in adolescent girls, categorized as complications of pregnancy, and obstetric and gynecological effects, are immature pelvic structures of pregnant teenagers that could cause cephalo-pelvic disproportion which could injure pelvic structures, thereby causing postpartum bleeding [44], hypertensive disorders of pregnancy and obstructed labor among teenagers [24], the likelihood of fistula experience and postpartum hemorrhage [43], and cesarean sections as a result of the mother’s medical conditions [14].

### 3.4. Unsafe Abortions

Abortions under unsafe circumstances were found to be a common health hazard for teenage mothers largely due to the illegality of abortion in the sub-region. Most abortions by adolescents were unsafely terminated [45], more adolescents indulged in unsafe abortions [32], abortions by adolescents were performed under unsafe circumstances [29], methods for abortions were obtained from non-formal providers [35], and crude methods such as insertion of objects into the vagina, a heavy message and the drinking of a herbal concoction were used [34].

### 3.5. Psychological Effects

The mental health and psychological effects related impact of adolescent pregnancy findings reported by the studies are sadness and an unhappy mood [37], moderate to severe depression, and major psychosocial effects due to pregnancy [28,31], such as feelings of fear, anger, shyness, and being miserable [30] were described. Findings of pregnancy-related stress [24], suicidal thoughts, ideations and feelings of rejection [33], fear, self-condemnation, guilt [22], and poorer coping ability and attitude toward pregnancy [41] were also found through this review. Lastly, three of the studies included in the review measured other outcomes different from the themes above. They reported not encouraging eating habits of pregnant adolescent women [27] and experiences of physical violence by pregnant adolescent girls [23,39].

## 4. Discussion

This systematic review was conducted to identify the adolescent maternal health impacts of pregnancy in three West African states. It found pregnant adolescent mothers to be anemic; they overly become exposed to pregnancy-related complications, they suffer from obstetric and gynecological effects, and they indulge in unsafe abortions, as well face pregnancy-related psychosocial stresses. These deleterious adverse health outcomes could have implications for women’s sexual and reproductive health and rights, as well as the attainment of SDG 3. All the studies that reported these deleterious medical effects that accompany adolescent pregnancy were conducted recently (post-MGDs), which is consistent with a study on the global burden of diseases that found adverse adolescent maternal outcomes as a substantial burden of adolescent morbidity and mortality in the Global South [4]. This review reported the health impacts of adolescent pregnancy from studies published in the last six years preceding this review, about six years into the start of the SDGs, and also in an era (twenty-first century) where the advancement in medical care is complemented by sophisticated information communication technology (ICT) driving the quality and efficacy of health care, bringing about an increment in quality and longer life span since the turn of the century [10]. However, over 70% of women of reproductive age face restrictive abortion laws in West Africa, where about 85% of abortions are performed in an unsafe manner [13]. The emergence and diffusion of Islam and Christianity together with traditional African religious beliefs in these countries make it challenging for the implementation of such scientific interventions, such as abortion care [46]. Some attitudes of healthcare providers influenced by religion, access and stigma, and victimization discourage women from patronizing the limited available safe abortion care services [47]. This forces many adolescent girls in the region to be in need of such services, as found through this study to engage in deadly abortion practices, where over 90% of the countries have prohibitive abortion rules [13]. Unsurprisingly, maternal mortality and morbidity indices in the selected countries are worrying [48]. These findings by this study are consistent with [49] stating that adolescent women are categorized within a special risk group of expectant mothers and routine antenatal care might not be enough to avert complications. What perhaps is not clear is the association of inadequate dietary intake, unhealthy eating habits, and intimate partner violence with adolescent pregnancy since only a couple of studies reported these outcomes. Young maternal age is found to be at an increased health risk during pregnancy [44]. Risks of complications and injuries such as obstetric fistulas, abortions, and even death could be a consequence of pregnancy in young women (10–19) [12]. The adverse health outcomes emanating from adolescent pregnancy are considered a public health hazard contributing significantly to maternal and child morbidity and mortality, and have been reported widely [8]. The health effects, as stated above, could manifest from the period of conception and may last a lifetime.

As has been revealed through this study, pregnancy could severely and adversely impact the health of girls in West Africa. One such adverse health outcome is anemia. Pregnancy during this period can contribute substantially to making them anemic without effective interventions. Anemia during pregnancy is due to physiological changes resulting in hemodilution and the increased requirements of iron [50]. A cross-sectional retrospective quantitative study found 68.9% of parturient adolescents to be anemic [51]. Cognitive performance is thought to be affected by anemia, in particular iron deficiency anemia. Iron deficiency anemia is a common phenomenon in adolescent pregnancy and iron treatment is required for a positive maternal outcome [52]. Anemic pregnant adolescents are found to have a postpartum hemorrhage, preeclampsia, and heart failure. The underlying factors for these adverse maternal outcomes are adolescent pregnancy itself [53], thus emphasizing the need to deal with the root cause which is adolescent pregnancy. Pregnancy-related complications, and obstetric and gynecological effects are also known to cause various degrees of harm to the health of adolescents. Pregnancy-induced hypertension and cesarean sections are commonly associated with adolescent pregnancy [54]. This is considered not only risky for adolescent childbearing but a risk factor for later pregnancies as well [45]. The development of obstetric fistulas among adolescent mothers accounted for up to 86% of treated cases with the majority of them from Sub-Saharan Africa in a clinical review study [55].

Another serious health impact is mental state disturbances. Postpartum depression in adolescent mothers has been reported [56]. In a systematic review in Sub-Saharan Africa and globally of adolescent girls and young women’s psychosocial experiences post-abortion, the prevalence of shame and abandonment was reportedly experienced by adolescents of all ages [57]. This population subgroup at this critical developmental stage is noted for its fragility in terms of mental health stability [56] and, therefore, pregnancy should be avoided if undesired, especially among the younger adolescents who are less than fifteen years old and who are known to suffer gravely to prevent it from becoming a precursor to mental health disorders. Reduced uptake and the use of contraception and reproductive health-related services among adolescents is a painful reality. The goal of SDG 3, target 3.7, for wider coverage of sexual and reproductive healthcare services for adolescents, truly deserves some more attention as contraception usage is regarded as the surest way of preventing unintended pregnancy in this population sub-group [32]. This is even more so as low coverage and patronage of this essential service is noted as a major determining factor in the incidence of adolescent pregnancy [11]. Restrictions on reproductive services to adolescents could be infringing on international conventions that are seeking to abolish all forms of discrimination, inequalities, and abusive practices against women [58]. International treaties such as The Convention on the Elimination of All Forms of Discrimination Against Women (CEDAW) and The African Charter on the Human and Peoples’ Rights and Rights of Women of Africa (Maputo Protocol) are some specific treaties adopted to ensure the protection of women’s sexual and reproductive rights. CEDAW, also referred to as the international bill of rights of women, calls on countries to guarantee adolescents access to reproductive health services and family planning. Similarly, the Maputo Protocol aims to guarantee sexual and reproductive health including safe abortion care (Article 14). Early marriages (under 18 years) which contribute to adolescent pregnancy are discouraged by the protocol. It further calls upon duty bearers in Article 14 to ensure a woman’s right to health, including respecting and promoting their sexual and reproductive health [59]. This suggests that much more needs to be performed to promote the uptake and usage of contraception by adolescents. Key to the expansion of coverage and uptake of contraception is education and information dissemination. Adopting and comprehensively implementing the WHO’s guidelines on adolescent sexual and reproductive health and rights [6] can help push the wheels of progress to attain the above target 3.7 of SDG 3, as sexual and reproductive health and right is a critical tool for fulfilling developmental goals [58]. Ensuring universal coverage of sexual and reproductive healthcare, as far as the adolescent girl population in these countries and to a large extend West Africa is concerned, could help build their capacity to take charge of their sexual live and significantly consolidate gains in the realization of SDG 3 [60]. HRBA to sexual reproductive health services and rights could promote increased access and uptake in women, particularly adolescents [58].

The study has some strengths and limitations. The study identified adverse adolescent maternal outcomes attributable to pregnancy in Ghana, Liberia, and Nigeria in West Africa through a comprehensive search and the religious application of the PRISMA 2020 guidelines [16]. The inclusion of DHS improves the quality of the review as it provides reliable and timely information on population and health in developing countries. However, the main sources of data were scientific databases and DHS; therefore, there is a possibility of missing out on some studies during the search process as they might not have published their findings in journals in these databases as adolescent pregnancy is a multidisciplinary topic. Some included studies had small sample sizes. Only studies in the English language published from 2016 to 2021 were used for this study. Contrary to the initial plan, meta-analyses are not performed as a result of a lack of reported statistically estimated effect sizes by most of the studies, thus making the study lack statistical power. In addition, there are no studies from all the West African countries which could limit this review’s generalization. 

## 5. Conclusions

In conclusion, the review identified anemia, complications of pregnancy, obstetric and gynecological effects, unsafe abortions, and mental health-related effects as negatively impacting the health and well-being of adolescent girls which could substantially contribute to maternal and child morbidity and mortality. As women suffer these ailments, and some sadly lose their lives rather prematurely through pregnancy as a result of restrictions on sexual and reproductive health and rights and safe abortion care which might be required exclusively by women, could suggest the need to prioritize HRBA to sexual and reproductive health issues in Ghana, Liberia, and Nigeria in West Africa. This unfortunate situation seems to infringe on their rights to health and ripples to hinder their full participation, and could expose adolescent girls to gender-based violence (GBV), exclusions, and inequities. Sexual reproductive health and rights services for adolescents should be scaled up across these countries and the entirety of the sub-region to help accelerate the attainment of SDG 3. Women’s rights to exclusive services, such as safe abortion care, should be regarded as a safe scientific intervention to promote health and well-being and, therefore, legal instruments, religion, and culture should not be used to discourage the provision and accessibility of such services. 

## Figures and Tables

**Figure 1 ijerph-20-00605-f001:**
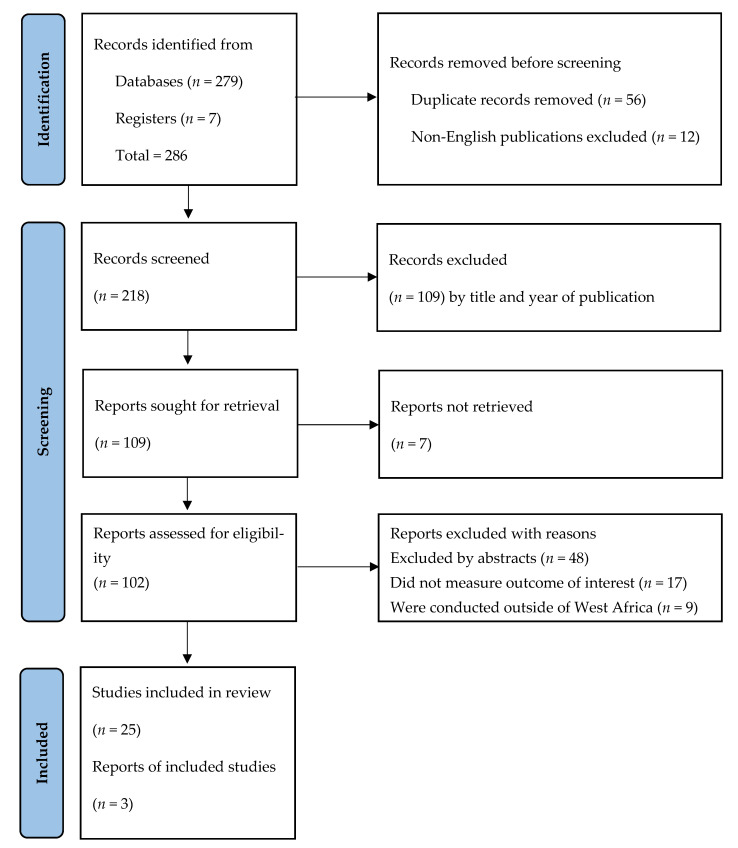
PRISMA flowchart of search results.

**Table 2 ijerph-20-00605-t002:** Main effects of adolescent pregnancy.

Title	Setting and Country	Type of Study	Impact of Pregnancy on Maternal Adolescent Health
Complications of pregnancy among adolescents and adult mothers treated in a public hospital, the republic of Liberia: a retrospective comparative study	Jackson Doe Referral Hospital, Liberia	Retrospective study	The most prevalent complication of pregnancy among teenagers was anemia.
Factors associated with iron deficiency anemia among pregnant teenagers in Ashanti Region, Ghana: a hospital-based prospective cohort study	29 communities in Kumasi Metropolis, Ghana	Prospective cohort study	Iron deficiency anemia is common among the pregnant teenagers studied.
Obstetric outcomes: a comparison of teenagers and adults in the Cape Coast Metropolis, Ghana	3 hospitals in Cape Coast Metropolis, Ghana	Observational study	Immature pelvic structures of pregnant teenagers could cause cephalo-pelvic disproportion, which would lead to injury to the pelvic structures, thereby causing bleeding after delivery
Beyond love: a qualitative analysis of factors associated with teenage pregnancy among young women with pregnancy experience in Bolgatanga, Ghana	Bolgatanga municipality, Northern Ghana	Qualitative study	Saddened or unhappy.
Comparative analysis of trends and determinants of anemia between adult and teenage pregnant women in two rural districts of Ghana	7 district health centers in the Ashanti region, Ghana	Retrospective study	The teenage group was found as more anemic.
The long-term effects of adolescent pregnancies in a community in Northern Ghana on subsequent pregnancies and births of the young mothers	Community-based, Ghana	Cross-sectional study	The findings of this study have shown that women with adolescent pregnancies experience more abortions.
Outcome of teenage pregnancy in a low resource setting: a comparative study	Federal teaching hospital, Ido, Ekiti, Nigeria	Retrospective study	Pregnancy-related complications such as hypertensive disorders of pregnancy, cephalo-pelvic disproportion/obstructed labor, and anemia were found among teenagers.
Association between adolescent motherhood and maternal and child health indices in Maiduguri, Nigeria: a community-based cross-sectional study	Maiduguri, capital of Borno state, Nigeria/	Cross-sectional study	Adolescent mothers were more likely to experience fistula and to have a postpartum hemorrhage.
Causative factors for sexual and reproductive health status of pregnant adolescent girls in urban communities of Lagos, Nigeria	Lagos Island, Nigeria/	Mixed (qualitative and quantitative)	The presence of morbidity in the form of anemia.
Sexual and reproductive health challenges of adolescent males and females in some communities of Plateau State, Nigeria	2 local government areas of Plateau State, Nigeria	Exploratory qualitative	Unsafe abortion.
Psychosocial effects of pregnancy on teenage mothers in Angwan Rukuba community, Jos, Plateau State, Nigeria	Angwan Rukuba, Jos, Plateau State, Nigeria	Descriptive study	Depression and substance abuse were the major psychosocial effects of pregnancy among respondents.
Nutritional knowledge and dietary intake habits among pregnant adolescents attending antenatal care clinics in urban community in Ghana	Ledzorkuku-Krowor in Greater Accra, Ghana	Cross-sectional study	The eating habits of adolescent pregnant women were not encouraging.
Decision-making preferences and risk factors regarding early adolescent pregnancy in Ghana: stakeholders’ and adolescents’ perspectives from a vignette-based qualitative study	Jamestown, Accra, Ghana	Qualitative study	Feelings ranged from, fear, anger, disappointment, frustration, misery, regret, and being shy.
Stress and resilience among pregnant teenagers in Ile-Ife, Nigeria	Ile-Ife, Osun State, Nigeria	Cross-sectional study	Most of the respondents were categorized as having a moderate level of perceived pregnancy-related stress.
To keep or not to keep? Decision making in adolescent pregnancies in Jamestown, Ghana	Jamestown Accra, Ghana	Qualitative/semi-structured in-depth interview	Adolescents who had abortion experiences were carried out under unsafe circumstances.
Contraceptive and abortion practices of young Ghanaian women aged 15–24: evidence from a nationally representative survey	Household-based, Ghana	National survey	Over half of young women used abortion methods obtained from non-formal providers.
Teenage pregnancy and experience of physical violence among women aged 15–19 years in five African countries: analysis of complex survey data	Nigeria	Survey	Physical violence among pregnant adolescents was five times higher compared to those who were not pregnant.
Descriptive epidemiology of anemia among pregnant women initiating antenatal care in rural Northern Ghana	Navarongo War Memorial Hospital, Ghana	Cross-sectional study	Expectant mothers less than 20 years old were more likely to be anemic.
Early motherhood: voices from female adolescents in the Hohoe Municipality, Ghana—a qualitative study utilizing Schlossberg’s Transition Theory	Hohoe municipality, Ghana	Qualitative study	Suicidal thoughts after pregnancy confirmation and the feeling of rejection by family and friends.
Exploring differences between adolescents and adults with perinatal depression-data from the expanding care for women with perinatal depression trial in Nigeria	Ibadan, Southwest Nigeria	Cluster randomized controlled trial	Adolescents had major depression compared with adults. Adolescents had significantly poorer adjustment and attitudes to pregnancy.
Perceived social support and depression among pregnant and child-rearing teenagers in Ile-Ife, Southwest Nigeria	Ile Ife, a community in Osun State, Southwest Nigeria	Descriptive study	Adolescents were categorized as having mild mood disturbance, experiencing borderline clinical depression, having moderate depression, and some were categorized as severely depressed.
“It’s like being involved in a car crash”: teen pregnancy narratives of adolescents and young adults in Jos, Nigeria	Jos, Plateau State, Nigeria	Qualitative study	Emotions described included fear, self-condemnation, and guilt about shaming their family.
Ghana Special Maternal Health Survey 2017	National, Ghana	Demographic and Health Survey	Drinking milk/coffee/alcohol/other liquid with sugar, drinking a herbal concoction, drinking another home remedy, using a herbal enema, inserting a substance into the vagina, heavy massage, excessive physical activity, tablets (exact kind unknown), and others.
Demographic and Health Survey 2019–2020	National, Liberia	Demographic and Health Survey	Experienced physical violence during pregnancy.
Nigeria Demographic and Health Survey, 2018	National, Nigeria	Demographic and Health Survey	Mothers’ medical conditions resulting in cesarean sections.

## Data Availability

Not applicable.

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
