# Peer review of "The Effects of Pregnancy: A Systematic Review of Adolescent Pregnancy in Ghana, Liberia, and Nigeria"

_ijerph, 2022, doi:10.3390/ijerph20010605_

Round 1
Reviewer 1 Report
Although the method used to conduct the systematic review are well described and the results adequately reported, I find some significant drawbacks that the authors must resolve.
Title: delete "effects of pregnancy" because it is repetitive.
Abstract: In this section, it is established that "The high adolescent pregnancy in West Africa is not without consequences ", then the study is not justified.
Introduction: the rationale for restricting the review to West Africa during a reduced period (2016-2021) child clearly established. Although the authors comment that in this region the rate of adolescent pregnancy is higher, this argument is not enough. For example, what characteristics does this region have that findings from other places can be extrapolated? or why findings from this region can be useful for other contexts? Why the last five years (2016-2021) can be different from the previous ones? Indeed, the authors describe previous research that shows the very same results that they reported herein. So, what is the novelty of this piece?
Results. In this section, the authors should report estimates that give support to their discussion and conclusions. For example, one comment in this section is "a common complication that occurs 246 among pregnant adolescents in these countries compared to other age groups of pregnant 247 women ". However, none figures or estimates are included to make sense of the phrase "compared to". Another example is the phrase "adolescents experienced more abortions", but the estimates are required to make clear the comparison. Please, included the estimates of each group/comparisons.
Discussion. Because in the results section the estimations are not reported, then in this section is not justified used phrases such as "risk group" or "higher risk".
The comments about SDG should be reduced because is not part of the aims of the review.
The comments about safe abortion are repetitive.
Finally, I found some sentences hard to understand. I underlined those in red (see the attached file).

Author Response
Dear Reviewer 1,
Thank you for giving us the opportunity to submit a revised manuscript titled “Effects of Pregnancy: A Systematic Review of the Health Impacts of Adolescent Pregnancy in West Africa” to International Journal of Environmental Research and Public Health. We appreciate the time and effort that the reviewer have dedicated to providing the valuable feedback on our manuscript. We are grateful to the reviewer for their insightful comments on our manuscript. We have been able to incorporate changes to reflect most of the suggestions provided by the reviewer. We have highlighted the changes within the revised manuscript.
Here is a point-by-point response to the reviewer’ comments and concerns.
Comments from Reviewer 1
- Comment 1: Title: delete "effects of pregnancy" because it is repetitive.
Response: Respectively agree and the title is rewritten in lines 2-5 (page 1).
- Comment 2: Abstract: In this section, it is established that "The high adolescent pregnancy in West Africa is not without consequences ", then the study is not justified.
Response: We agree to this comment and have rephrased in lines 12-14, 20-21, 25 (page 1). Further justification of the study is in lines 146-158 (page 4).
- Comment 3: Introduction: the rationale for restricting the review to West Africa during a reduced period (2016-2021) child clearly established. Although the authors comment that in this region the rate of adolescent pregnancy is higher, this argument is not enough. For example, what characteristics does this region have that findings from other places can be extrapolated? or why findings from this region can be useful for other contexts? Why the last five years (2016-2021) can be different from the previous ones? Indeed, the authors describe previous research that shows the very same results that they reported herein. So, what is the novelty of this piece?
Response: Respectively, though it might not be novel, but it brings together for the first time the health impacts on the adolescent girl through a review in the region. Lines 146-158 (page 4) gives more detail justification. The reasons for the 2016- 2021 year are also found in lines 152-154 (page 4).
- Comment 4: Results. In this section, the authors should report estimates that give support to their discussion and conclusions. For example, one comment in this section is "a common complication that occurs 246 among pregnant adolescents in these countries compared to other age groups of pregnant 247 women ". However, none figures or estimates are included to make sense of the phrase "compared to". Another example is the phrase "adolescents experienced more abortions", but the estimates are required to make clear the comparison. Please, included the estimates of each group/comparisons. Discussion. Because in the results section the estimations are not reported, then in this section is not justified used phrases such as "risk group" or "higher risk".
Response: We totally agree with the above concerns, however, the estimates as extracted from the studies have been summarized in table 2 (page 8-12). The data were analyzed qualitatively using SWiMS (narrative synthesis) in Lines 220-225 (page 5). The lack of statistical power has been acknowledged as a weakness of the study in lines 644-646(page 22). The comparative phrases have been removed in lines 339-341(page 16), page 17 line 412.
- Comment 5: The comments about SDG should be reduced because is not part of the aims of the review.
Response: This is a valid concern, but we may have not made it clear enough. More information has been added in lines 150-154 (page 4) as part of the objective for the review.
- Comment 6: The comments about safe abortion are repetitive.
Response: Thanks for this kind observation. Some parts have been expunged in lines 430 (page 18). Page 19 line 502 also removed.
- Comment 7: Finally, I found some sentences hard to understand. I underlined those in red (see the attached file).
Response: We agree that they could have been better presented. These have been rephrased in lines 85-130 (page 2-3), page 4 -5 lines 176-184, page 5 line 218-223, page 7 lines 271-277, page 8 lines 241-255, page 20 lines 553-554, and page 21 line 636-639.
Reviewer 2 Report
Manuscript ID: ijerph-2050673
Full Title: Effects of Pregnancy: A Systematic Review of the Health Impacts of Adolescent Pregnancy in West Africa
Thank you for your submission.
Adolescent pregnancy is a major public health problem, particularly in Africa. Hence, the global health impacts of adolescent pregnancy, specifically in regions of the developing world, are a significant chronic public health concern. For that reason, any publication aiming to highlight aspects of this important issue is appreciated.
In this paper, a systematic review was conducted to identify the adolescent maternal health impacts of pregnancy in three selected West African states.
However, in reviewing the paper, I noticed some issues that should be addressed before this review might be considered for publication:
Title and Keywords
The systematic review refers to adolescent pregnancy in three countries: Ghana, Liberia, and Nigeria. Indeed, these countries are part of West Africa, but this review did not aim to review all research on the suggested issue throughout West Africa. I think the authors should change the title accordingly, specifically, “Effects of Pregnancy: A Systematic Review of the Health Impacts of Adolescent Pregnancy in Ghana, Liberia, and Nigeria” (or similar). In that case, “West Africa” can be used as a keyword instead of the names of the three countries.
Introduction
The Introduction’s structure and flow are good.
· The authors mention that all three selected countries are “English speaking countries in West Africa since publications in English language only were considered for the study” (lines 69–70). Are these the only English-speaking countries in West Africa? Was it the single criterion for choosing these three? I think that more should be written about choosing only these three. Readers should understand the logic of this decision. Furthermore, I suggest that after explaining the decision, the authors be consistent in the information they provide the readers about each country (e.g., there is historical information about Liberia and Ghana but not Nigeria). The country information should be written in a way that would expose readers from all over the globe to relevant details on the topic reviewed.
· A short explanation should be added regarding “Sustainable Development Goal (SDG) 3” (lines 83–84). The explanation should include information regarding specific subsections connected to the review aims (for example, 3.1. Maternal Mortality).
Materials and Methods
· Selection criteria: “Publications that discussed adolescent pregnancies in the selected countries were also used” (lines 103–104). Used together with what? What do the authors mean by “were also used”?
“Publications that are not related to adolescent pregnancy” (line 110). This criterion is kind of obvious, isn’t it?
Inclusion and exclusion criteria are a major part of systematic reviews and should be written systematically and very clearly. This paragraph (2.2; lines 100–116) should be reorganized.
· More explanation regarding the SWIM is needed.
Results
· The first four lines of this section repeat information already reported under the “Data extraction and quality assessment” section, but this section is missing other information. Please add the missing information: How many studies were not in consensus? What was the reliability between reviewers?
· Why are the results of the “The Joanna Briggs appraisal tool” not presented?
· How was the quality of the studies evaluated, and what did you find?
· 3.1. Characteristics of included studies: An overall description that does not mention the references for each statement is not enough. For example: “There were 3 retrospective studies” (line 182) – Which ones? Or another example: “One of the two longitudinal studies used Adolescent Birth Outcomes, Ghana (ANBOG)” (lines 183–184). To which study does this statement relate?
· “Their pregnancies were desired and undesired. Some of the pregnancies were carried to term, birthed, whilst others were aborted” (lines 192–193). Please add the results (percentages/ratios): How many pregnancies were desired? Undesired? Some of the pregnancies were carried to term (How many?), birthed (How many?), whilst others were aborted (How many?)
· Tables 1 and 2 should be merged into one table to help readers follow the information (using landscape orientation should help). The title of each study does not need to be presented in the table. Instead, the studies can be numbered the Authors of each study could be linked to the References list.
· The “Impact of pregnancy on maternal adolescent health” (Table 2) should be presented in the table, as one of the four main themes was found. It should be written in the table much more concisely and clearer. The detailed description is repeated in sections 3.2.–3.5 and is unnecessary in the table. Furthermore, please present the information without unneeded description. For example, “The most prevalent complication of pregnancy among teenagers was anemia” should be just “Anemia.”
Discussion
The Discussion section should be shorter and less repetitive in the presentation of the ideas, making the sequence unclear and confusing. Overall, I suggest you shorten and restructure the Discussion according to the study’s objectives.
First, the maternal health impacts of adolescent pregnancy in the three countries should be discussed. This should be the main part of the Discussion because, according to the authors, this was the review’s primary aim. Only then should the authors discuss other issues (adopting more pragmatic approaches to check unplanned pregnancies, help in achieving Sustainable Development Goal 3, the legal restriction, etc.). This part of the Discussion needs to be much shorter and organized in a way that each important idea is addressed but written only once, clear, and straightforward.
Furthermore, reconsider the context in which issues such as limited access to safe abortion services and the emergence and diffusion of Islam and Christianity with traditional African religious beliefs that make implementing scientific interventions like abortion care challenging are presented. None of these issues are specific to adolescent pregnancy. Unfortunately, these facts are a reality for women of all reproductive ages, including adolescents, in almost every West African county. I believe these ideas should appear in this review and that it is vital they be discussed—but that discussion should be in the bigger context of health issues affecting adolescent pregnancy.
References
Several references (e.g., 7, 11, 37) lack a DOI address.
Author Response
Dear Reviewer 2,
Thank you for giving us the opportunity to submit a revised manuscript titled “Effects of Pregnancy: A Systematic Review of the Health Impacts of Adolescent Pregnancy in West Africa” to International Journal of Environmental Research and Public Health. We appreciate the time and effort that the reviewer have dedicated to providing your valuable feedback on our manuscript. We are grateful to the reviewer for their insightful comments on our manuscript. We have been able to incorporate changes to reflect most of the suggestions provided by the reviewer. We have highlighted the changes within the revised manuscript.
Here is a point-by-point response to the reviewer’ comments and concerns.
Comments from Reviewer 2
- Comment 1: Title and Keywords The systematic review refers to adolescent pregnancy in three countries:Ghana, Liberia, and Nigeria. Indeed, these countries are part of West Africa, but this review did not aim to review all research on the suggested issue throughout West Africa. I think the authors should change the title accordingly, specifically, “Effects of Pregnancy: A Systematic Review of the Health Impacts of Adolescent Pregnancy in Ghana, Liberia, and Nigeria” (or similar). In that case, “West Africa” can be used as a keyword instead of the names of the three countries.
Response: Respectively agree and the title rewritten in (page 1) lines 2-5. West Africa included as keyword in line 25(page 1).
- Comment 2: Introduction The Introduction’s structure and flow are good. The authors mention that all three selected countries are “English speaking countries in West Africa since publications in English language only were considered for the study” (lines 69–70). Are these the only English-speaking countries in West Africa? Was it the single criterion for choosing these three? I think that more should be written about choosing only these three.
Response: Respectively agree as more information will enrich the background. More information has been added in lines 137-144 (page 4) additional information in the broader context of West Africa.
- Comment 3: Readers should understand the logic of this decision. Furthermore, I suggest that after explaining the decision, the authors be consistent in the information they provide the readers about each country (e.g., there is historical information about Liberia and Ghana but not Nigeria). The country information should be written in a way that would expose readers from all over the globe to relevant details on the topic reviewed.
Response: Respectively agree as more information will enrich the background. More information has been added in lines 107-115. Also line 133-144 (page 4) gives additional information in the broader context of West Africa.
- Comment 4: A short explanation should be added regarding “Sustainable Development Goal (SDG) 3” (lines 83–84). The explanation should include information regarding specific subsections connected to the review aims (for example, 3.1. Maternal Mortality).
Response: We respectively agree on the need for more information. This has been added in lines 150-157 (page 4).
- Comment 5: Materials and Methods Selection criteria: “Publications that discussed adolescent pregnancies in the selected countries were also used” (lines 103–104). Used together with what? What do the authors mean by “were also used”?
Response: Respectively agree as this is an important part of a review. This has been rewritten/reorganized in lines 162-174 (page 4).
- Comment 6: “Publications that are not related to adolescent pregnancy” (line 110). This criterion is kind of obvious, isn’t it?
Response: Yes, we agree and thus removed and rewritten/reorganised in lines 162-174 (page 4).
- Comment 7: Inclusion and exclusion criteria are a major part of systematic reviews and should be written systematically and very clearly. This paragraph (2.2; lines 100–116) should be reorganized.
Response: Totally agree to this critical observation and has been rewritten/reorganised in lines 147-159 (page 4).
- Comment 8: More explanation regarding the SWIM is needed.
Response: Very valid input. Further explanation has been provided in lines 220-223 (page 5).
- Comment 9: Results The first four lines of this section repeat information already reported under the “Data extraction and quality assessment” section, but this section is missing other information. Please add the missing information: How many studies were not in consensus? What was the reliability between reviewers? Why are the results of the “The Joanna Briggs appraisal tool” not presented? How was the quality of the studies evaluated, and what did you find?
Response: Respectively agree to this concern. The Joanna Briggs appraisal tool was independently used by two reviewers to ascertain their quality for inclusion or exclusion. Overall 28 publications were included. 3 studies were excluded upon resolution with the third reviewer after initial disagreements between two. These are contained in lines 176-213, page 4-5.
- Comment 10: 1. Characteristics of included studies: An overall description that does not mention the references for each statement is not enough. For example: “There were 3 retrospective studies” (line 182) – Which ones? Or another example: “One of the two longitudinal studies used Adolescent Birth Outcomes, Ghana (ANBOG)” (lines 183–184). To which study does this statement relate?
Response: We totally agree for the need for identifying the characteristics with their respective studies. 3.1 (page 6) lines 271-285 gives an overview of the study characteristics with more detailed information summarized in Table 1 with their respective studies (page 7-11).
- Comment 11: “Their pregnancies were desired and undesired. Some of the pregnancies were carried to term, birthed, whilst others were aborted” (lines 192–193). Please add the results (percentages/ratios): How many pregnancies were desired? Undesired? Some of the pregnancies were carried to term (How many?), birthed (How many?), whilst others were aborted (How many?).
Response: We totally agree with the above concerns, however, the estimates as extracted from the studies have been summarized in Table 2 (page 11-16). The data were analyzed qualitatively using SWiMS (narrative synthesis) as indicated in lines 216-228 (page 5). The lack of statistical power has been acknowledged as a weakness of the study in lines 644-646 (page 22).
- Comment 12: Tables 1 and 2 should be merged into one table to help readers follow the information (using landscape orientation should help). The title of each study does not need to be presented in the table. Instead, the studies can be numbered the Authors of each study could be linked to the References list.
Response: This is a good suggestion but merging them will make it more difficulty to present a detailed feature of the studies used. Using two tables to the best of our knowledge allow for a more detailed information on the included studies (Table 1) (page 8-11) and Table 2 also presents the summary of the extracted impact estimates in pages 11-16.We also suggest that the two tables be formatted in landscape orientation for easy reference.
- Comment 13: The “Impact of pregnancy on maternal adolescent health” (Table 2) should be presented in the table, as one of the four main themes was found. It should be written in the table much more concisely and clearer. The detailed description is repeated in sections 3.2.–3.5and is unnecessary in the table. Furthermore, please present the information without unneeded description. For example, “The most prevalent complication of pregnancy among teenagers was anemia” should be just “Anemia”.
Response: This is a good suggestion; however, it presents impact estimates in qualitative summary as extracted from the studies. The main themes as grouped are presented in pages 16 and 19, lines 337-530.
- Comment 14: Discussion The Discussion section should be shorter and less repetitive in the presentation of the ideas, making the sequence unclear and confusing. Overall, I suggest you shorten and restructure the Discussion according to the study’s objectives.
Response: This is a good suggestion, and the entire discussion segment has been rewritten/reorganised incorporating these useful comments in pages 19-22, lines 532-639.
- Comment 15: First, the maternal health impacts of adolescent pregnancy in the three countries should be discussed. This should be the main part of the Discussion because, according to the authors, this was the review’s primary aim. Only then should the authors discuss other issues (adopting more pragmatic approaches to check unplanned pregnancies, help in achieving Sustainable Development Goal 3, the legal restriction, etc.). This part of the Discussion needs to be much shorter and organized in a way that each important idea is addressed but written only once, clear, and straightforward.
Response: This is a good suggestion, and the entire discussion segment has been rewritten/reorganised incorporating these useful comments in pages 19-22, lines 532-639.
- Comment 16: Furthermore, reconsider the context in which issues such as limited access to safe abortion services and the emergence and diffusion of Islam and Christianity with traditional African religious beliefs that make implementing scientific interventions like abortion care challenging are presented. None of these issues arespecific to adolescent pregnancy. Unfortunately, these facts are a reality for women of all reproductive ages, including adolescents, in almost every West African county. I believe these ideas should appear in this review and that it is vital they be discussed—but that discussion should be in the bigger context of health issues affecting adolescent pregnancy.
Response: We respectively agree as these inputs will enrich our work. The entire discussion segment has been rewritten/reorganised incorporating these useful comments in pages 19-22 with reduced lenth.
- Comment 17: References Several references (e.g., 7, 11, 37) lack a DOI address.
Response: This is a good observation. This however was an oversight which have been added as suggested in the reference section, pages 25-29, lines 845-1003,
Round 2
Reviewer 2 Report
Accept in present form. Good-Luck!
Author Response
Dear Reviewer,
We appreciate the time and effort that you have dedicated to providing your valuable feedback on our manuscript.
Thank you to your relentless support.
Sincerely yours,
Authors